# Impact of COVID-19 Disease on the Development of Osteomyelitis of Jaws: A Systematic Review

**DOI:** 10.3390/jcm13154290

**Published:** 2024-07-23

**Authors:** Emmanouil Vardas, Daniela Adamo, Federica Canfora, Maria Kouri, Konstantina Delli, Michele Davide Mignogna, Nikolaos Nikitakis

**Affiliations:** 1Department of Oral Medicine and Pathology and Hospital Dentistry, National and Kapodistrian University of Athens, 15772 Athens, Greece; mbardas@gmail.com (E.V.); kourimari@yahoo.gr (M.K.); nnikitakis1@yahoo.com (N.N.); 2Department of Neuroscience, Reproductive Sciences, and Dentistry, University of Naples “Federico II”, 80138 Naples, Italy; federica.canfora@unina.it (F.C.); mignogna@unina.it (M.D.M.); 3Department of Oral Diseases and Oral and Maxillofacial Surgery, University of Groningen, 9712 CP Groningen, The Netherlands; k.delli@umcg.nl

**Keywords:** osteomyelitis, jaw, necrotizing osteomyelitis, COVID-19, maxillofacial, mandible

## Abstract

**Background/Objectives:** Osteomyelitis is characterized by an inflammatory process affecting both bone and bone marrow, leading to cell death and the formation of bone sequestrum. Recent literature from the past five years has documented instances of osteomyelitis following infections of SARS-CoV-2. This systematic review explores the link between osteomyelitis of the jaw (OMJ) and COVID-19 infections. **Methods:** This review adhered to the PRISMA guidelines, systematically analyzing literature from 2020 to 2024 sourced from databases including Medline, Embase, Scopus, and Web of Science. PROSPERO ID: CRD42024526257. **Results:** The review selected 42 articles, detailing 201 cases of osteomyelitis of the jaw related to COVID-19 (COMJ). The demographic breakdown included 195 male (74.4%) and 67 female patients (25.6%), with a median age of 52.7 years, ranging from 24 to 71 years. A significant portion of COMJ patients (41.5%) were hospitalized due to COVID-19, and 58.5% received corticosteroid therapy. Diabetes mellitus was a common comorbidity among COMJ patients (65.1%). Most cases involved maxilla (182 cases; 90.5%), with nearly half showing sinus involvement (49.4%). The mandible was affected in 19 cases (9.5%). Mucormycosis and aspergillosis emerged as the predominant fungal infections, identified in 103 (51.2%) and 50 (24.9%) cases, respectively. **Conclusions:** Individuals with pre-existing health conditions such as diabetes mellitus who have been treated for COVID-19 are at an increased risk of developing OMJ, particularly maxillary fungal osteomyelitis. COMJ poses a significant diagnostic and therapeutic challenge for dental and maxillofacial professionals, who are often the first to encounter these cases.

## 1. Introduction

On 11 March 2020, the World Health Organization (WHO) declared COVID-19 a pandemic caused by the SARS-CoV-2 virus [1]. Increasing evidence shows that COVID-19 complications can affect multiple body organs, including the oral cavity [2]. SARS-CoV-2 uses angiotensin-converting enzyme 2 (ACE2) receptors to enter human cells. These receptors are present in oral organs like the tongue and salivary glands, suggesting that the oral cavity plays a significant role in COVID-19 infection [3,4].

The downregulation of ACE2 receptors by the virus contributes to endothelial dysfunction and hyperinflammation, leading to endothelial cell inflammation and microvascular problems throughout various organs. This process shows how SARS-CoV-2 induces clotting through specific pathways and the interplay between thrombosis and inflammation [4].

The COVID-19 pandemic has significantly impacted oral healthcare delivery and the lives of dentists worldwide. Changes in clinical protocols and the emergence of specific oral pathologies related to COVID-19 are notable [5]. High ACE2 expression in oral epithelial cells makes the oral cavity particularly susceptible to SARS-CoV-2, leading to distinct oral symptoms associated with COVID-19 [6].

Oral complications in COVID-19 patients can result from factors such as improper medication use, weakened immunity, vascular damage, local and systemic inflammation, and neglect of oral hygiene during treatment [4]. Reported oral manifestations include mucosal ulcerative lesions, loss of smell, taste alterations, gingivitis, plaque on the tongue, inflammation of Wharton’s-duct papillae, temporomandibular disorders, xerostomia, halitosis, herpetic gingivostomatitis, fungal infections, and parotitis [7].

Moreover, a notable emerging complication in the maxillofacial area related to COVID-19 is osteomyelitis of the jaw (OMJ). The immune dysregulation and microvascular alterations coupled with prolonged hospitalization and invasive procedures in critically ill patients have predisposed individuals to secondary infections [8].

Osteomyelitis, an infection of the bone and bone marrow, is a serious condition that can lead to severe tissue damage characterized by high morbidity if not properly managed [9]. This infection, typically caused by bacteria but also by fungi or viruses, may arise through hematogenous spread, contiguous infection from adjacent tissues, or direct inoculation following trauma or surgical procedures [9]. While osteomyelitis can affect any bone, certain anatomical regions, such as the jaw, are particularly susceptible due to their unique anatomical and physiological characteristics. In the context of the jaw, the presence of teeth, periodontal pockets, and the pneumatization of the maxilla provide a fertile environment for anaerobic bacterial colonization, which can predispose individuals to osteomyelitis following dental procedures or maxillary sinus infections [10]. The mandible is the most commonly affected bone within the facial skeleton due to the different anatomy and less extensive blood supply of the maxilla, with osteomyelitis of the maxilla accounting for only 1% to 6% [10,11]. 

The Zurich classification system categorizes osteomyelitis into acute, secondary chronic, and primary chronic types [10,12]. Acute and secondary chronic OMJ are the same condition, distinguished by the arbitrary time limit of four weeks following the onset of symptoms [10]. 

Necrotizing osteomyelitis, a more severe variant, is characterized by extensive bone necrosis and rapid progression, often leading to significant complications if not promptly addressed [13,14]. This form of osteomyelitis is frequently seen in immunocompromised individuals, such as patients who have developed COVID-19 or those with multiple comorbidities, such as poorly controlled diabetes, which can exacerbate the severity of the infection and complicate treatment outcomes [15].

The conventional treatment of OMJ usually requires surgical debridement followed by long-term antibiotic therapy [16]. Despite advancements in antibiotics and dental care, OMJ continues to be a disease with a poor outcome that represents a complex challenge for healthcare providers [17]. 

This systematic literature review aims to comprehensively examine the current evidence on the association between COVID-19 and osteomyelitis, including also necrotizing forms. The review will explore the incidence, clinical presentations, diagnostic challenges, treatment strategies, and outcomes of osteomyelitis in the context of COVID-19. By synthesizing data from various studies and case reports, this review seeks to identify patterns and elucidate the potential mechanisms by which SARS-CoV-2 may influence the development and progression of osteomyelitis. Additionally, this review will highlight gaps in current knowledge and propose directions for future research to better understand the interplay between COVID-19 and bone infections, ultimately aiming to improve clinical management and patient outcomes in this complex clinical scenario.

## 2. Materials and Methods

For this systematic review, the examination and gathering of data followed the 2020 edition of the Preferred Reporting Items for Systematic Reviews and Meta-analyses (PRISMA) standards [18]. An extensive search was performed on 15 April 2024 across these databases: Web of Science, Medline, Embase, and Scopus. The study was registered on the PROSPERO platform (ID: CRD42024526257) (Figure 1).

In particular, based on the methods outlined, Hypothesis 0 (H0) is: there is no significant relationship between necrotizing osteomyelitis of the jaws and SARS-CoV-2 infection; Hypothesis 1 (H1) is: there is a significant relationship between necrotizing osteomyelitis of the jaws and SARS-CoV-2 infection.

### 2.1. Eligibility Criteria

The eligibility criteria targeted studies evaluating cases of OMJ related to COVID-19 (COMJ), restricting the scope to research published between 2020 and 2024 and articles in English. Exclusions were applied to: -articles describing patients with COVID-19 who experienced osteonecrosis of the jaw unrelated to the study focus;-articles involving patients with COVID-19 who were undergoing or had undergone treatment related to medications causing osteonecrosis of the jaw and subsequently developed jaw osteonecrosis (MRONJ).

There were no limitations on geographic location, patient age, or gender. Any articles published after 15 April 2024 were omitted. We actively excluded meta-analyses, systematic reviews, additional review articles, non-peer-reviewed articles, expert perspectives, published abstracts, articles in languages other than English, withdrawn articles, and studies relying solely on self-reported data.

### 2.2. Search Strategy and Keywords

This systematic review aimed to address the question: “Is there a relationship between necrotizing osteomyelitis of the jaws and SARS-CoV-2 infection?”. This objective guided the search strategy and inclusion criteria to focus on identifying relevant studies that explore the association between these specific conditions. 

To locate pertinent studies, a search strategy was developed that included a combination of medical subject headings (MeSH) and keywords relevant to the research question. The keywords were strategically chosen to cover a broad yet specific spectrum of topics including (1) COVID-19, (2) osteomyelitis, (3) fungal osteomyelitis, (4) jaws, (5) bone necrosis, and (6) maxillofacial.

These terms were used to search databases and filter the literature to include studies that specifically discussed the impact of SARS-CoV-2 infection on the development of OMJ. This approach ensured a comprehensive collection of data pertinent to the review’s objectives.

### 2.3. Screening Process

In adherence to the PRISMA guidelines, a meticulous screening process was employed to ensure thoroughness and accuracy. From 934 articles identified, duplicates were removed, leaving 451 articles for evaluation. Two independent reviewers evaluated these 451 articles, all of which were unanimously accepted for full-text review, following the predefined inclusion and exclusion criteria. A third reviewer was involved to resolve any disagreements between the two authors during the title and abstract screening, as well as the full-text screening. This streamlined approach facilitated a consistent application of the criteria throughout the review.

### 2.4. Data Extraction

The designated individual responsible for data management utilized a structured data extraction table created in Excel (Microsoft, 2023, Redmond, WA, USA) to systematically record and organize the gathered information from the selected studies to capture a comprehensive range of domains critical for evaluating the impact of COMJ.

The specific domains included authors, study type, year of publication, country, number of cases, gender, age, post-COVID-19 condition, COVID-19 therapy, vaccination status, hospitalization status, symptoms, onset of symptoms, location of infection, sinus involvement, presence of exposed bone, radiological evaluations, tests other than histopathologic evaluation, fungal infections, bacterial infections, tooth involvement, treatment modalities, comorbidities, medication intake, possible mechanisms of disease, and histopathologic examination.

This methodical approach allowed for a detailed and uniform collection of data across studies, facilitating subsequent analysis and interpretation of the findings related to the relationship between COVID-19 and osteomyelitis in the jaws.

### 2.5. Data Analysis

Data extracted from the included studies were analyzed using descriptive statistics and narrative synthesis. Descriptive statistics were employed to summarize quantitative data, such as the number of patients, age, and mortality rate. Narrative synthesis was used to describe and interpret the qualitative data, such as clinical presentation, diagnostic criteria, and treatment methods.

## 3. Results

### 3.1. Data Collection 

A total of 934 articles were retrieved collectively from Medline, Embase, Scopus, and Web of Science. After deleting 483 duplicates, 451 articles were screened by title and abstract, of which 122 records were assessed for eligibility. Full-text screening revealed 42 studies that were included in the systematic review. These 42 studies detailed 201 cases of COMJ (Figure 1).

### 3.2. Characteristics of Included Studies

Table 1 outlines data from multiple countries, summarizing the types of reports (case reports, case series, prospective studies, retrospective studies), number of reports, number of patients, and references. India was the most involved country, with 25 reports and 144 patients. This pattern of documentation is consistent across other countries listed, such as Iran, Uzbekistan, and Pakistan, with varying patient and report counts differentiated by type of study.

### 3.3. Patients’ Data

Table 2 provides a demographic and clinical profile of the patients, including gender distribution (71.7% male, 28.3% female) and median age (52.72 years). The prevalence of comorbidities was as follows: diabetes (65.1%), hypertension (19.8%), heart disease (2.8%), renal disease (2.4%), cancer (1.9%), anemia (1.4%), pneumonia (1.4%), hypothyroidism (1.4%), arthritis (0.9%), obesity (0.9%), and other conditions (3.3%). Only 5.7% of patients had no comorbidities, while 18.4% suffered from more than one systemic disease.

### 3.4. COVID-19-Related Data

Table 3 details information about COVID-19-related hospitalization, corticosteroid administration, and the timing of presentation to the dental center post-COVID-19-diagnosis. Specifically, 41.5% of the included patients were hospitalized due to COVID-19, 58.5% received corticosteroid therapy, and 22.2% received a diagnosis at the dental center more than 61 days after the onset of the disease.

### 3.5. Bone-Necrosis-Related Data

Table 4 documents symptoms at presentation, the site of necrosis, involvement of the sinus, and types of radiologic evaluations used. Specifically, pain and swelling were reported in 60.4% and 47.2% of cases, respectively. The maxilla was predominantly involved in 91.03% of the patients, while only 8.5% reported mandibular involvement. Regarding complications, 58.96% had sinus involvement, and 34.9% presented with exposed bone. Tooth involvement was frequent (57.6%), and CT was the most prescribed radiologic evaluation (CT: 28.3%; CECT: 24.5%; CBCT: 7.1%). The types of infections diagnosed were predominantly fungal (mucormycosis: 47.6%; Aspergillus: 25.9%) and less frequently bacterial (Actinomyces: 3.3%; Escherichia coli: 3.3%). Both fungal and bacterial infections were reported in 2.4% of cases, while 1.9% had no infection-related findings.

### 3.6. Treatment Data (Surgical and Non-Surgical)

Table 5 describes the treatment protocols. Regarding surgical treatments, maxillectomy was predominantly performed, both as total (20.9%) and subtotal maxillectomy (20.8%), followed by surgical debridement (25%). Among the non-surgical treatments, amphotericin B was prescribed in 27.8% of cases, followed by fluconazole (9.4%) and prednisolone (8.02%).

## 4. Discussion

This systematic review sought to elucidate the relationship between COVID-19 and OMJ by analyzing studies published between 2020 and 2024. By including 42 studies that detailed 201 cases of COMJ, the review provides a comprehensive understanding of this emerging complication.

The data collected from various countries revealed a significant concentration of studies in Asia (35 out of the 42 studies; 83.33%), particularly in India (25 out of 42; 59.52%).

The remaining studies were distributed as follows: America (4/42, 9.52%), Africa (2/42, 4.76%), and Europe (1/42, 2.39%) (Table 1). 

This geographic distribution likely reflects the lack of the use of broad-spectrum antibiotics in countries with underdeveloped health systems and low income levels and the higher prevalence of mucormycosis in India, estimated at 140 cases per million people, which is approximately 80 times higher than that in developed countries [19].

The demographic data revealed a higher prevalence of COMJ in males (71.7%) compared to females (28.3%), with a median age of 52.7 years, ranging from 8 to 83, with a male-to-female ratio of 2.5:1. This aligns with other studies indicating that males are more susceptible to severe COVID-19 outcomes, possibly due to differences in immune response and comorbidities such as diabetes and hypertension, which were highly prevalent in COMJ patients (65.1% and 19.8%, respectively). However, these findings deviate slightly from other studies in the literature, which report variations in the male-to-female ratio up to 5.2:1 and demonstrate a more even distribution of patients across various age groups [60,61].

A significant portion of COMJ patients (41.5%) were hospitalized due to COVID-19, and 58.5% received corticosteroid therapy. The use of corticosteroids, although beneficial for managing severe COVID-19, may contribute to COMJ by suppressing the immune system and increasing susceptibility to infections such as mucormycosis and aspergillosis, which were predominant in the reviewed cases (51.2% and 24.9%, respectively).

Although osteomyelitis of the maxilla is typically rare, due to the high vascularity of the maxilla, which provides significant collateral blood flow, and its porous nature, making it less susceptible to infections compared to the mandible [60,61], our data analysis showed a different trend. The maxilla was predominantly affected (91.03%), with a high incidence of sinus involvement (58.96%) and exposed bone (34.9%). Specifically, osteomyelitis of the maxilla was nearly 10 times more prevalent than osteomyelitis of the mandible (8.5%) in these cases. This increased prevalence may be attributed to the unique conditions created by the COVID-19 pandemic, such as the heightened incidence of fungal infections like mucormycosis, which have disproportionately affected the maxillary region.

Pain (60.4%) and swelling (47.2%) were the most common symptoms observed, followed by tooth mobility (39.2%) and pus discharge (32.1%), consistent with findings in the literature [20].

Orofacial pain is one of the most common reasons patients seek treatment, particularly in conditions such as inflammation and ulceration, which are present in COMJ. In these conditions, there is an enhanced expression and function of acid-sensing ion channels, leading to the depolarization of nerve axons and resulting in pain [62,63,64]. Acid-sensing ion channels (ASICs) are a group of proton-gated ion channels that play a critical role in the sensation of pain, especially in response to tissue acidosis (e.g., inflammation and ischemia) [65]. Studies have shown that increased expression of ASICs in the trigeminal nerve, which innervates the orofacial region, can lead to heightened pain perception [66,67]. Moreover, chronic inflammation, a hallmark of osteomyelitis, can further sensitize these ion channels, exacerbating pain. This is particularly relevant in COMJ, where persistent inflammatory stimuli can lead to continuous activation of ASICs, contributing to ongoing pain and discomfort [68]. The presence of ulceration can also stimulate nociceptors, intensifying pain signals to the brain [69].

These symptoms highlight the need for healthcare providers to consider COMJ in the differential diagnoses for post-COVID-19 patients presenting with such issues. Additionally, the use of local anesthetics, anti-inflammatory medications, and anti-epileptics should be considered in conjunction with other treatments for COMJ [70]. 

Radiological evaluations, including CT (28.3%), CECT (24.5%), and CBCT (7.1%), played a crucial role in diagnosing COMJ. The high usage of CT scans underscores their importance in detecting bone necrosis and sinus involvement, which are critical for planning appropriate treatment strategies. This reliance on radiological tools emphasizes the need for prompt and accurate imaging to facilitate early diagnosis and intervention.

Diabetes mellitus was a common comorbidity among COMJ patients (65.1%), followed by hypertension (19.8%). All other comorbidities were reported in less than 3% of cases (Table 2). 

Although the role of diabetes mellitus in the pathogenesis of osteomyelitis requires further investigation, there is a strong correlation between diabetes mellitus and the incidence of osteomyelitis [60]. Hyperglycemia in diabetic patients is associated with impaired immune responses and disrupted wound healing, making them more susceptible to bacterial infections.

Additionally, a significant percentage of COVID-19 patients (58.5%) received high doses of corticosteroid therapy. The extensive use of corticosteroids during COVID-19 treatment can also lead to immune suppression and induce permanent hyperglycemia, even in previously non-diabetic patients, further increasing the risk of fungal osteomyelitis.

It is important to note that no studies have compared non-diabetic COVID-19 patients who did not receive steroids and developed mucormycosis with those who did receive steroids and developed mucormycosis. Consequently, establishing a definitive cause-and-effect relationship between COVID-19, mucormycosis, and corticosteroid use remains challenging [71].

Many articles in the literature report that OMJ is typically polymicrobial, involving organisms such as Actinomyces, Streptococcus, Bacteroides, Peptostreptococcus, Candida, and other opportunistic pathogens [65,66,67]. During the maturation of the infectious process, there is often a shift in the predominant flora. However, our review indicates a notable deviation from this pattern, showing a predominance of fungal infections (81%), with mucormycosis being the most common infection (47.6% of all infections). Bacterial infections were present in only 15.1% of the cases.

This shift towards fungal predominance can be attributed to several factors related to the COVID-19 pandemic. The widespread use of broad-spectrum antibiotics in COVID-19 treatment can disrupt normal microbial flora, creating an environment more conducive to fungal overgrowth. Additionally, the use of oxygen masks, particularly in severely and critically ill patients, may introduce or exacerbate fungal infections due to the humid and enclosed environment they create [34].

Furthermore, the extensive use of corticosteroids for managing severe COVID-19 cases impairs the immune system, making patients more susceptible to opportunistic infections such as mucormycosis [72]. This is particularly true for patients with underlying conditions like diabetes mellitus, which was highly prevalent among the COMJ cases reviewed. Hyperglycemia in diabetic patients further compromises the immune response and wound healing processes, thereby increasing the likelihood of developing severe fungal infections [73].

The first symptoms of COMJ developed an average of 100.99 days (range, 0–620 days) after the diagnosis of COVID-19 and presentation at the dental center. In this cohort, 41.5% of patients were hospitalized due to COVID-19, 3.8% did not require hospitalization, and data for 54.7% of patients were missing. These findings highlight the extended latency period between COVID-19 diagnosis and the onset of COMJ symptoms, which is significantly longer compared to other reports in the literature.

A particularly relevant comparison can be drawn from a systematic review and meta-analysis conducted by Özbek et al. [68], which analyzed 958 cases of COVID-19-associated mucormycosis. The authors reported that the first symptoms of mucormycosis developed, on average, 22.3 days (range, 0–240 days) after the diagnosis of COVID-19. This considerable difference in the onset of symptoms suggests that COMJ may have a more prolonged incubation period or that other contributing factors, such as underlying comorbidities or the severity of the initial COVID-19 infection, could influence the timing of symptom development.

Moreover, this disparity underscores the need for ongoing vigilance and long-term monitoring of patients who have recovered from COVID-19, especially those with pre-existing conditions like diabetes or those who have received corticosteroid treatment. 

Furthermore, it is essential to consider the various factors that might contribute to this prolonged latency. For instance, extended treatments with corticosteroids and antibiotics may mask the initial symptoms of COMJ, thereby delaying diagnosis. Additionally, variations in treatment protocols across different healthcare facilities and countries can influence the timing of symptom onset.

Although no studies have specifically focused on the variants of COVID-19 in connection with the development of COMJ, this represents a crucial area of interest. There may be significant differences in how various virus variants affect patients’ susceptibility to secondary infections such as mucormycosis. Therefore, future research should concentrate on analyzing the relationship between COVID-19 variants and the incidence of COMJ to identify any patterns and risk factors associated with each variant.

The extended latency period observed in this study indicates that healthcare providers should maintain a high index of suspicion for COMJ well beyond the acute phase of COVID-19 recovery. This approach can facilitate early diagnosis and intervention, potentially improving patient outcomes by addressing complications before progression to more severe stages.

Regarding the treatment of COMJ, protocols varied and included both surgical interventions and non-surgical treatments. Specifically, the surgical interventions reported in the studied cases were total maxillectomy (20.9%), subtotal maxillectomy (20.8%), and surgical debridement (25%). Non-surgical treatments primarily involved antifungal medications such as amphotericin B (27.8%) and fluconazole (9.4%), along with corticosteroids (8.02%).

The significant reliance on surgical treatments underscores the severity of COMJ and the necessity for aggressive management to prevent further complications. This approach is crucial given the invasive and progressive nature of osteomyelitis associated with COVID-19, which can rapidly compromise vital craniofacial structures. Early and decisive treatment strategies are therefore essential to mitigate the severe outcomes associated with this condition.

### Limitations

Despite the strengths of this systematic review, there are notable limitations that should be acknowledged.

The literature search was not exhaustive, as it did not encompass all existing databases. This could result in the omission of relevant studies that might report additional cases of COMJ. Expanding the search to include more databases would provide a more complete picture and reduce the risk of missing critical data.

A significant portion of the included studies were case reports (23/42, 54.77%) and case series (16/42, 38.09%). These types of studies are generally considered to provide a lower level of evidence compared to randomized controlled trials or large cohort studies. The reliance on case reports and series can limit the generalizability and robustness of the findings. Future reviews should aim to include more high-quality studies to strengthen the evidence base.

A notable limitation of our study is the absence of data regarding comorbidities and medication administration in the included articles. This lack of comprehensive data prevented us from accurately including and correlating these factors in our analysis. Consequently, the potential influence of comorbidities and medication use on the incidence of osteomyelitis following COVID-19 could not be fully assessed, which may impact the overall findings and conclusions of our study.

It is important to note that the etiology of osteomyelitis typically requires a bacterial or fungal infection, which often originates from an odontogenic or oral cavity infection. Consequently, distinguishing between different sources of infection, such as periodontitis—a serious gum infection caused by bacterial accumulation that can damage bones and teeth as it progresses—was not feasible. The screened studies did not consistently exclude the possibility of an odontogenic etiology for osteomyelitis, making it challenging to isolate hygienic-lifestyle-behavior-related issues. This aspect underscores the complexity of determining the exact origin of the infections leading to osteomyelitis in the context of this review.

The geographical concentration of studies in Asia, particularly India, may introduce a regional bias. This could affect the generalizability of the findings to other regions with different healthcare systems and epidemiological profiles. Efforts should be made to include studies from a more diverse range of countries to provide a more global perspective.

The review included only one retrospective study and one prospective study of case series. These study designs are crucial for understanding the temporal relationship and causality between COVID-19 and COMJ. An increase in the number of such studies would provide more reliable evidence regarding the progression and outcomes of COMJ.

Comparative studies are scarce in the existing literature. Comparative studies are essential for evaluating the effectiveness of different treatment modalities and understanding the impact of various risk factors on the development and progression of COMJ. Future research should focus on conducting comparative studies to fill this gap.

A significant portion of the data, especially regarding corticosteroid intake and hospitalization status, was missing (35.4% and 54.7%, respectively). Missing data can lead to biased conclusions and affect the validity of the findings. Ensuring more comprehensive data collection in future studies is necessary to improve the reliability of the results.

## 5. Conclusions

This systematic review elucidates the complex relationship between COVID-19 and OMJ, highlighting several key findings and implications for clinical practice and future research. Most studies on COMJ originated from Asia, particularly India, reflecting the higher prevalence of mucormycosis in regions with underdeveloped healthcare systems.

The demographic data revealed a higher prevalence of COMJ in males, with a median age of 52.7 years, highlighting the importance of considering regional and demographic factors when evaluating and managing COMJ.

Treatment protocols varied, with a significant reliance on surgical interventions, highlighting the severity of COMJ and the need for aggressive management.

Ongoing vigilance and long-term monitoring of post-COVID-19 patients, especially those with pre-existing conditions and those who received corticosteroid treatment, are essential. Future research should focus on understanding the impact of different COVID-19 variants on COMJ, optimizing treatment protocols, and exploring preventive strategies. Moreover, a new systematic review correlating the incidence of avascular necrosis with medication use in patients who had COVID-19 could be a valuable next step.

Developing standardized treatment protocols and preventive measures, including antifungal prophylaxis for high-risk patients, is essential. Collaboration among healthcare professionals is vital to address the multifaceted challenges of COMJ and improve patient outcomes.

## Figures and Tables

**Figure 1 jcm-13-04290-f001:**
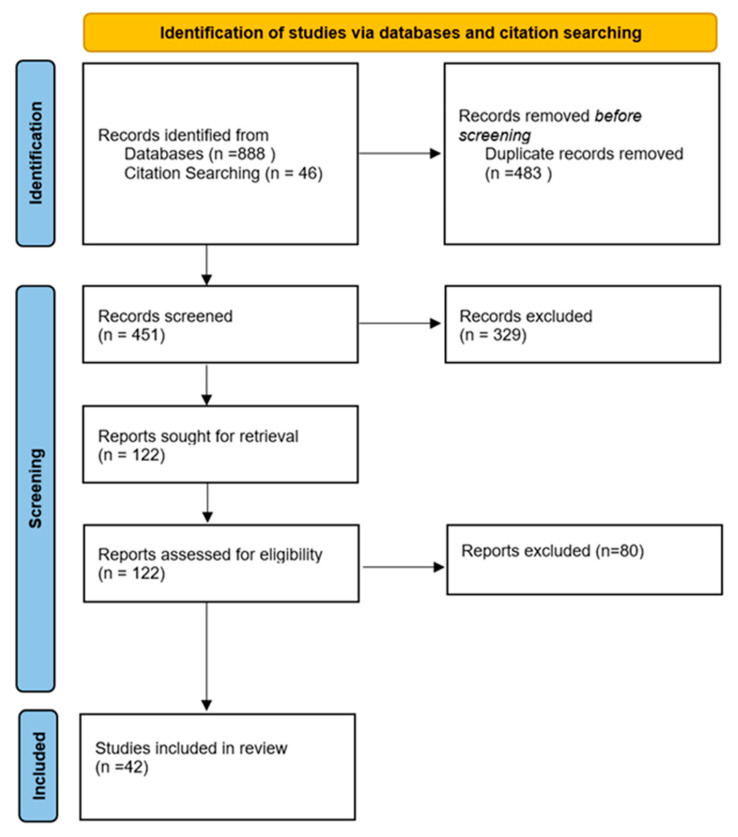
PRISMA 2020 flow diagram.

**Table 1 jcm-13-04290-t001:** Characteristics of the included studies.

Country	Number of Reports/ Number of Patients	Kind of Reports/ Authors [ref] (Number of Patients)	Number of Reports	Number of Patients	References
*India*	25/144	Case report: Durugkar et al. (1); Protyusha et al. (1); Vasanthi et al. [19] (1); Sihmar et al. (1); Indira et al. (1); Arafat et al. [20] (1); Ambereen et al. (1); Jawanda et al. (1); Gombra et al. (1); George et al. (1); Bhanumurthy et al. (1); Arewar et al. (1); Shah et al. (1)	13	13	[21,22,23,24,25,26,27,28,29,30,31,32,33]
		Case series: Ansari et al. (3); Paavai et al. (2); Sai Krishna et al. (2); Kunusoth et al. [34] (3); Pranave et al. (7); Wadde et al. (3); Prajwal et al. (2); Shirke et al. (4); Sodhi et al. (6)	9	32	[35,36,37,38,39,40,41,42,43]
		Prospective study: Suresh et al. (39); Datarkar et al. (47)	2	86	[44,45]
		Retrospective study: Khan et al. (13)	1	13	[2]
*Iran*	5/18	Case report: Moaddabi et al. (1); Arian et al. (1); Khoshkhou et al. (1)	3	3	[46,47,48]
		Case series: Grillo et al. (2), Motevasseli et al. (3)	2	15	[49,50]
*Uzbekistan*	2/5	Case report: Boymuradov et al. (1)	1	1	[51]
		Case series: Boymuradov et al. (4)	1	4	[19]
*Pakistan*	1/1	Case report: Arshad et al. [32] (1)	1	1	[52]
*Brazil*	2/2	Case report: Riva et al. (1); Pauli et al. (1)	2	2	[20,53]
*Mexico*	2/21	Case report: López-González et al. (1)	1	1	[54]
		Case series: Urias-Barreras et al. (20)	1	20	[55]
*Egypt*	2/16	Case series: El Charkawi et al. (2); Said Ahmed et al. (14)	2	16	[56,57]
*Texas*, *USA*	1/1	Case report: Arora et al. (1)	1	1	[58]
*Korea*	1/1	Case report: Kang et al. (1)	1	1	[59]
*Croatia*	1/3	Case series: Kvolik Pavić et al. (3)	1	3	[8]
TOTAL	42/212				

**Table 2 jcm-13-04290-t002:** Patient demographics and comorbidities.

Parameters	Mean (Range)	Number (%)
**Gender**		
Male		152 (71.7)
Female		60 (28.3)
**Age (years)**		
Total	52.72 (8–83)	
Male	51.79 (8–78)	
Female	55.04 (24–83)	
**Comorbidities**		
Diabetes		138 (65.1)
Hypertension		42 (19.8)
Heart disease		6 (2.8)
Renal disease		5 (2.4)
Cancer		4 (1.9)
Anemia		3 (1.4)
Pneumonia		3 (1.4)
Hypothyroidism		3 (1.4)
Arthritis		2 (0.9)
Obesity		2 (0.9)
Other		7 (3.3)
No comorbidities		12 (5.7)
More than one comorbidity		39 (18.4)
NA		45 (22.2)

**Table 3 jcm-13-04290-t003:** COVID-19 hospitalization and treatment data.

Parameters	Number (%)	Mean (Range)
**COVID-19 hospitalization**		
Yes	88 (41.5)	
No	8 (3.8)	
Data unavailable	116 (54.7)	
**Corticosteroid administration for COVID-19 therapy**		
Yes	124 (58.5)	
No	13 (6.1)	
Data unavailable	75 (35.4)	
**Days after COVID-19 diagnosis (at presentation to the dental center)**		100.99 (0–630)
Less than 30 days	44 (20.8)	
Between 31 and 60 days	73 (34.4)	
More than 61 days	47 (22.2)	

**Table 4 jcm-13-04290-t004:** Bone-necrosis-related data.

Parameters	Number (%)
**Complaint (at presentation to the dental center)**	
Pain	128 (60.4)
Swelling	100 (47.2)
Tooth mobility	83 (39.2)
Pus discharge	68 (32.1)
Paresthesia	41 (19.3)
Ulceration	26 12.3)
Exposed bone	19 (8.96)
Halitosis	16 (7.5)
Tenderness	4 (1.9)
Headache	4 (1.9)
Discomfort	3 (1.4)
Other	17 (8.01)
**Site of necrosis**	
Maxilla	193 (91.03)
Mandible	18 (8.5)
Both	1 (0.5)
**Sinus involvement**	125 (58.96)
**Exposed bone**	74 (34.9)
Yes	122 (57.6)
No	25 (11.8)
NA	65 (30.7)
**Radiologic evaluation**	
CT	60 (28.3)
CECT	52 (24.5)
CBCT	15 (7.1)
MSCT	3 (1.4)
MDCT	1 (0.5)
MRI	11 (5.2)
OPG	10 (4.7)
Scintigraphy	1 (0.5)
Unspecified	1 (0.5)
More than one	19 (8.96)
NA	110 (51.9)
**Fungal infection**	
Mucormycosis	101 (47.6)
Aspergillus	55 (25.9)
Mixed fungal	8 (3.8)
Candida	3 (1.4)
Curvularia	2 (0.9)
Not specified	3 (1.4)
**Bacterial infection**	
Actinomyces	7 (3.3)
Escherichia coli	7 (3.3)
Mixed bacterial	4 (1.9)
Klebsiella	3 (1.4)
Staphylococcus	2 (0.9)
Streptococcus	2 (0.9)
Acinetobacter	1 (0.5)
Pseudomonas	1 (0.5)
Enterococcus	1 (0.5)
Prevotella	1 (0.5)
Proteus	1 (0.5)
Not specified	2 (0.9)
**Mixed fungal and bacterial**	5 (2.4)
**No infection**	4 (1.9)
**NA**	5 (2.4)

**Abbreviations:** MDCT: multidetector computed tomography; MSCT: multislice computed tomography; CECT: contrast-enhanced computed tomography; CT: computed tomography; CBCT: cone beam computed tomography; ACT: axial computed tomography; MRI: magnetic resonance imaging; OPG: orthopantomography; NA: not available.

**Table 5 jcm-13-04290-t005:** Treatment data.

Parameters	Number (%)
**Surgical treatment**	
Maxillectomy	44 (20.9)
Subtotal maxillectomy	43 (20.8)
Surgical debridement	53 (25.0)
Sequestrectomy	14 (6.6)
Teeth extraction	13 (6.1)
FESS	11 (5.2)
Eye exenteration	6 (2.8)
Alveolectomy	4 (1.9)
**Non-surgical treatment**	
Amphotericin B	59 (27.8)
Fluconazole	20 (9.4)
Prednisolone	17 (8.02)
Hyperbaric oxygen	15 (7.1)
Posaconazole	9 (4.2)
Clindamycin	5 (2.4)
Dexamethasone	8 (3.8)
Metronidazole	2 (0.9)
Gentamycin	2 (0.9)
**Combined treatment**	114 (53.82)

**Abbreviations:** FESS: functional endoscopic sinus surgery.

## Data Availability

Data available on request from the authors.

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
