# Peer review of "Impact of COVID-19 Disease on the Development of Osteomyelitis of Jaws: A Systematic Review"

_jcm, 2024, doi:10.3390/jcm13154290_

Round 1

Reviewer 1 Report

Comments and Suggestions for Authors

The authors of the systematic review titled "Impact of COVID-19 disease on the development of osteomyelitis of jaws: a systematic review 2020-2024." aimed to explore patterns and elucidate the potential mechanisms by which SARS-CoV-2 may influence the development and progression of osteomyelitis.

The Introduction section gives an overview of the impact of COVID-19 infection on oral health, noting that osteomyelitis can be a serious complication related to COVID-19. I suggest adding what previous systematic articles already found on this topic and what is still undisclosed.

It is necessary to add hypothesis 0 (H0) and hypothesis 1 (H1) in the Materials and Methods part. Also, list all inclusion criteria and operators that you applied.

Please add Figure 1 in the text of the Results (3.1. Data collection). 

The Discussion is fairly well-written, with major results explained. 

Conclusion should be shortened and only highlight the most important findings of the study, not repeat almost everything presented in the Results section.

Comments on the Quality of English Language

Minor English editing is required.

Author Response

The authors of the systematic review titled "Impact of COVID-19 disease on the development of osteomyelitis of jaws: a systematic review 2020-2024." aimed to explore patterns and elucidate the potential mechanisms by which SARS-CoV-2 may influence the development and progression of osteomyelitis.

  1. The Introduction section gives an overview of the impact of COVID-19 infection on oral health, noting that osteomyelitis can be a serious complication related to COVID-19. I suggest adding what previous systematic articles already found on this topic and what is still undisclosed.

Thank you for your valuable suggestion. We appreciate your insight and we have incorporated it into our manuscript. In the revised Introduction section, we have added a summary of findings from previous articles on the impact of COVID-19 on oral health and osteomyelitis. Additionally, we have highlighted the areas that remain undisclosed, emphasizing the gaps in current research that our study aims to address.

  1. It is necessary to add hypothesis 0 (H0) and hypothesis 1 (H1) in the Materials and Methods part. Also, list all inclusion criteria and operators that you applied.

Thank you for your comment. We have added this part in the Materials and Methods section: “In particular, based on the methods outlined, the hypothesis 0 (H0) is: there is no significant relationship between necrotizing osteomyelitis of the jaws and SARS-CoV-2 infection; the hypothesis 1 (H1) is: there is a significant relationship between necrotizing osteomyelitis of the jaws and SARS-CoV-2 infection.”

  1. Please add Figure 1 in the text of the Results (3.1. Data collection). 

Thank you for your comment. We have added the figure in the Results section.

  1. The Discussion is fairly well-written, with major results explained. 

Thank you for your comment.

  1. Conclusion should be shortened and only highlight the most important findings of the study, not repeat almost everything presented in the Results section.

Thank you for your comment. We have shortened the conclusion section accordingly.

Reviewer 2 Report

Comments and Suggestions for Authors

Dear Authors,

Thank you very much for taking the time and effort to conduct this essential study that presents the iceberg phenomenon in Oral and Maxillofacial Surgery. During the coronavirus disease (COVID-19) pandemic, many authors witnessed a changing pattern of dental and maxillofacial pathologies. This systematic review (2020-2024) evaluated the effect of COVID-19 (SARS-CoV-2) on the osteomyelitis of the jaws. The main results of the study showed that patients with pre-existing health conditions and treated for COVID-19 are at an increased risk of developing osteomyelitis. From an academic standpoint, I believe this study will improve our understanding of the association between COVID-19 and maxillofacial hard-tissue infections. However, I still have some questions.

1. Please remove "2020-2024" from the title and add it to the Material and Methods Section.

2. Medication-related osteonecrosis of the jaw (MRONJ) remains poorly understood, which generates difficulty in performing a differential diagnosis between MORNJ and Osteomyelitis of the Jaw; both pathologies have similar clinical patterns, but different histories. In this systemic review, why did you not extract a more comprehensive list of medications used to treat SARS-CoV-2 and comorbidities?

3. Did all the screened studies exclude the possible odontogenic etiology of osteomyelitis? Doesn’t this include hygienic lifestyle behavior-related issues?

4. Please extend the limitations of the study.

Best Wishes!

Reviewer

Comments on the Quality of English Language

Moderate editing of English language required.

Author Response

Thank you very much for taking the time and effort to conduct this essential study that presents the iceberg phenomenon in Oral and Maxillofacial Surgery. During the coronavirus disease (COVID-19) pandemic, many authors witnessed a changing pattern of dental and maxillofacial pathologies. This systematic review (2020-2024) evaluated the effect of COVID-19 (SARS-CoV-2) on the osteomyelitis of the jaws. The main results of the study showed that patients with pre-existing health conditions and treated for COVID-19 are at an increased risk of developing osteomyelitis. From an academic standpoint, I believe this study will improve our understanding of the association between COVID-19 and maxillofacial hard-tissue infections. However, I still have some questions.

  1. Please remove "2020-2024" from the title and add it to the Material and Methods Section.

Thank you for pointing this out. We agree to remove "2020-2024" from the title and add it to the Material and Methods section.

  1. Medication-related osteonecrosis of the jaw (MRONJ) remains poorly understood, which generates difficulty in performing a differential diagnosis between MORNJ and Osteomyelitis of the Jaw; both pathologies have similar clinical patterns, but different histories. In this systemic review, why did you not extract a more comprehensive list of medications used to treat SARS-CoV-2 and comorbidities?

Thank you for pointing this out. In many cases, there was missing data about medication administration. Additionally, we believe that a new systematic review correlating the incidence of avascular necrosis with medication related to patients who had COVID-19 could be a further step. We have added this aspect to the limitations section and the conclusion. In particular, we have included “A notable limitation of our study is the presence of missing data regarding comorbidities and medication administration in the included articles. This lack of comprehensive data prevented us from accurately including and correlating these factors in our analysis. Consequently, the poten-tial influence of comorbidities and medication use on the incidence of osteomyelitis following COVID-19 could not be fully assessed, which may impact the overall findings and conclusions of our study”, while in the conclusions: “Moreover, a new systematic review correlating the incidence of avascular necrosis with medica-tion use in patients who had COVID-19 could be a valuable next step.”

  1. Did all the screened studies exclude the possible odontogenic etiology of osteomyelitis? Doesn’t this include hygienic lifestyle behavior-related issues?

Thank you for your insightful comment. It was not possible to explore the specific aspects of odontogenic and hygienic behavior within this systematic review due to limitations in the available data from the included studies. However, we have added this aspect into the limitations section “it is important to note that the etiology of osteomyelitis typically requires a bacterial or fungal infection, which often originates from an odontogenic or oral cavity infection. Consequently, distinguishing between different sources of infection, such as periodontitis—a serious gum infection caused by bacterial accumulation that can damage bones and teeth as it progresses—was not feasible.The screened studies did not consistently exclude the possibility of an odontogenic etiology for osteomyelitis, making it challenging to isolate hygienic lifestyle behavior-related issues. This aspect underscores the complexity of determining the exact origin of the infections leading to osteomyelitis in the context of this review.”

  1. Please extend the limitations of the study.

Thank you for your comment. We have implemented this section accordingly.

Reviewer 3 Report

Comments and Suggestions for Authors

It is not a new topic, unfortunately there is a lack of support, it is a controversial topic, there are different factors due to which osteomyelitis can occur and not exactly exclusively due to COVID

Author Response

It is not a new topic, unfortunately there is a lack of support, it is a controversial topic, there are different factors due to which osteomyelitis can occur and not exactly exclusively due to COVID

Dear reviewer,

Thank you for your comment. We agree that osteomyelitis can result from various factors, not exclusively due to COVID-19. Our study does not suggest that osteomyelitis occurs solely because of COVID-19. The aim of our research was to investigate the increased number of cases of fungal and bacterial osteomyelitis in the maxillofacial region following Coronavirus Disease 2019, as reported in recent literature. To the best of our knowledge, this is the first systematic review correlating the increased incidence of necrotizing osteomyelitis to COVID-19, although many case reports and case series exist in the literature.
